# The Impact of Culinary Processing, including Sous-Vide, on Polyphenols, Vitamin C Content and Antioxidant Status in Selected Vegetables—Methods and Results: A Critical Review

**DOI:** 10.3390/foods12112121

**Published:** 2023-05-24

**Authors:** Grzegorz Kosewski, Magdalena Kowalówka, Sławomira Drzymała-Czyż, Juliusz Przysławski

**Affiliations:** Chair and Department of Bromatology, Faculty of Pharmacy, Poznan University of Medical Sciences, ul. Rokietnicka 3, 60-806 Poznań, Poland; grzegorzkosewski@ump.edu.pl (G.K.); mkowalowka@ump.edu.pl (M.K.); jprzysla@ump.edu.pl (J.P.)

**Keywords:** plants, technological processes, vacuum cooking, TAS, ABTS, DPPH, FRAP, TP, Vit. C

## Abstract

This study presents various research methods and results analysis of the total antioxidant status (TAS), polyphenols content (PC) and vitamin C content in selected plant materials (vegetables) subjected to various technological processes, including sous-vide. The analysis included 22 vegetables (cauliflower white rose, romanesco type cauliflower, broccoli, grelo, col cabdell cv. pastoret, col lllombarda cv. pastoret, brussels sprouts, kale cv. crispa–leaves, kale cv. crispa–stem, toscana black cabbage, artichokes, green beans, asparagus, pumpkin, green peas, carrot, root parsley, brown teff, white teff, white cardoon stalks, red cardoon stalks and spinach) from 18 research papers published in 2017 to 2022. The results after processing by various methods such as conventional, steaming and sous-vide cooking were compared to the raw vegetable results. The antioxidant status was mainly determined by the radical DPPH, ABTS and FRAP methods, the polyphenol content by the Folin–Ciocalteu reagent and the vitamin C content using dichlorophenolindophenol and liquid chromatography methods. The study results were very diverse, but in most studies, the cooking techniques contributed to reducing TAS, PC and vitamin C content, with the sous-vide process being most beneficial. However, future studies should focus on vegetables for which discrepancies in the results were noted depending on the author, as well as lack of clarity regarding the analytical methods used, e.g., cauliflower white rose or broccoli.

## 1. Introduction

Vegetables can be eaten raw or cooked and are a valuable source of nutrients including vitamins and polyphenols with proven antioxidant properties [1]. Antioxidants present in vegetables can scavenge reactive oxygen species (ROS) which contribute to oxidative stress, a major cause of many diseases and accelerates the ageing process. Phenolic compounds and vitamin C have anti-cancer, anti-ageing, neuroprotective and cardioprotective effects [2,3].

However, traditional thermal processing methods including classic cooking, stewing, baking and steaming cause changes in the chemical composition (mainly the content of secondary metabolites) and organoleptic properties of vegetables, causing the loss of nutrients, as well as contributing to oxidation processes [4,5,6]. The modern sous-vide method (French under vacuum) involves cooking raw materials in a vacuum-sealed thermostable, plastic bag in a water bath or convection steam oven at a strictly controlled temperature not exceeding 100 °C for a longer time than classical methods [7,8,9,10]. The vacuum seal prevents valuable nutrients from entering the decoction by osmosis and reduces the oxidation of phenolic compounds [11,12]. The slow heat treatment also protects against the loss of moisture and volatile compounds, so vegetables acquire a delicate and juicy structure and retain their intense taste, aroma and colour [8,12,13,14]. This process also reduces the loss of vitamins and other nutrients [14], and the lack of oxygen prevents lipid oxidation [15].

Raw vegetables do not always have the highest antioxidant potential or polyphenol content compared to processed vegetables, as an elevated temperature and other factors may result in greater bioavailability of the active ingredients [16]. The sous-vide method increases the bioavailability of vitamins, minerals and phytochemicals [16] and may reduce the loss of vitamin C compared to conventional cooking methods [17]. To date, most previous studies focused on animal products subjected to the sous-vide process, with research on the nutritional value of plant products beginning only a few years ago [18]. 

## 2. Methods and Search Strategy

This study systematically analysed the studies regarding the total antioxidant status, polyphenols and vitamin C content of selected vegetables (22 vegetables) subjected to conventional cooking (CC), steaming (S) and sous-vide (SV). For comparison, the study results were expressed as percentages, i.e., the percentage increase/decrease in the antioxidant properties and polyphenols and vitamin C content to assess differences between cooked and raw vegetables. Based on the inclusion criteria, 18 articles were selected for cluster analysis (Figure 1).

## 3. Analytical Methods Used for the Antioxidant Potential and Polyphenols and Vitamin C Content

The most frequently used method to determine the antioxidant activity (AA) of various vegetables was the DPPH free radical reduction method [19,20,21,22,23,24,25,26], which involves the spectrometric measurement of the absorbance of a methanolic DPPH solution with a vegetable extract in relation to the control sample, i.e., a pure methanol solution with a radical at a wavelength of 515 nm (Table 1) [27,28]. The second most commonly used method was the cation radical ABTS [29,30,31], which also involves the spectrophotometric measurement of absorbance at a wavelength of 734 nm [32,33]. The less frequently used method was the reduction of iron ions (Fe^3+^ to Fe^2+^) or FRAP [20,26,29]. All these spectrophotometric methods are based on the SET mechanism, i.e., the transfer of a single electron [34]. In the DPPH radical and the ABTS methods, the most common solvent used to prepare the vegetable extracts was 80% methanol and 70% methanol for the FRAP method. The selection of an appropriate solvent for the extraction of antioxidant substances is important to maximise the extraction of active ingredients [35]. AA results for DPPH and ABTS methods were most often presented in terms of Trolox (vitamin E equivalent) (Table 1).

Determining the amount and type of polyphenols in vegetables allows for a broader analysis of their antioxidant potential. The most common method used to determine the polyphenols content in vegetables was the spectrophotometric method (λ = 756) using the Folin–Ciocalteu reagent (F–C). In an alkaline environment, the acids contained in F-C are reduced and the oxidised polyphenols give a blue colour [36,37]. High-Pressure Liquid Chromatography (HPLC) [38,39] using C18 columns (150 mm × 4.6 mm, 2.6 µm or 3 × 75 mm, 2.7 µm) [31] and a prodigy column (5 μm ODS3 100 A, 250 × 4.60 mm) [23], DAD detector, gradient flow, or HPLC-MS/MS using a C18 column (C3 × 75 mm, 2.7 μm), gradient and isocratic flow, ion spray voltage set at 3500 V [40] were used for more accurate analysis and separation of individual polyphenols Typically, the polyphenols were extracted in 70% [19,20] or 80% [24,25,33,34,35,36,37,38,39] methanol with the addition of formic acid [40] and 90% [21]. Some studies used 50% [41] or 80% ethanol [42]. One extraction method involved the use of water with sodium hydroxide and hydrochloric acid to determine the bound polyphenolic phase (Table 1) [30]. 

The antioxidant properties of vegetables were also determined by vitamins including β-carotene, folic acid and vitamin C [43], the content of which can be determined by spectrophotometric, titration, enzymatic, chemiluminometric, fluorometric, amperometric methods, as well as gas chromatography (GC) and liquid chromatography (LC) methods [44]. HPLC and UPLC (Ultra HPLC) methods are more accurate, sensitive, shorter and more efficient than other methods, particularly for the determination of low concentrations of vitamin C [45,46]. The most commonly used solvent to determine the vitamin C content was water with the addition of metaphosphoric acid [19,20,47] or acetic acid [41]. The most popular methods for vitamin C determination were the spectrophotometric method using dichlorophenolindophenol with the addition of oxalic acid [24,41,47] and HPLC with the DAD detector (isocratic flow of 1 mL/min), where the mobile phase was most often diluted sulphuric acid (Table 1) [19,20,21]. 

**Table 1 foods-12-02121-t001:** Conditions of sous-vide method, extractions and determination methods total antioxidant status and the content of polyphenols and vitamin C in vegetables.

Conditions of Sous-Vide Method		Extraction	TAS	TP	Vit. C	References
Temp.	Time		Methods			
90 °C	45–50 min	TAS and TP: first stage (fs): 0.16 mol/dm^3^ HCL in 80% methanol, second stage: supernatant from fs re-extracted on 70% acetone	ABTS	Folin–Ciocalteu reagent	Megazyne assay procedure K-ASCO 01/14	[29]
84 °C	30–60 min	TAS: methanol/water (80:20)	ABTS FRAP	Folin–Ciocalteu reagent	-	[26]
80 °C	15 min, 90 min	TP, DPPH, FRAP: 70% methanol Vit. C: 20 mL solution with 30 g/L meta-phosphoric acid and 80 mL/L acetic acid	FRAP DPPH	Folin–Ciocalteu reagent	HPLC: LC 18 column (250 × 4.6 mm, 5 μm), isocratic solvent system, (0.1 mL/min of sulphuric acid), flow rate 1 mL/min, UV-VIS photodiode array detector (254 nm)	[19,20]
90 °C	5, 10, 15 min	TP and TAS: 90% methanol	DPPH	Folin–Ciocalteu reagent	HPLC: column Coregel 87H3 (7.8 × 300 mm), isocratic elution, flow rate 1 mL/min, mobile phase—0.02 N H_2_SO_4_, detector—photo diode array (SPD-M20A) 254 nm	[21]
90 °C	50 min	TP, DPPH, FRAP: 80% methanol	ABTS FRAP	Folin–Ciocalteu reagent	-	[30]
90 °C, and 100 °C/90 °C	35, 45, 55 min and 25, 30, 35 min	TP: methanol/water (80:20) TAS: methanol/water (50:50)	DPPH	Folin–Ciocalteu reagent	-	[22]
80 °C	15 min	Vit. C: aqueous with 1% metaphosphoric acid	-	-	Spectrophotometry with 2,6-dichlorophenolindo-phenol (λ = 515 nm)	[46]
90 °C	30 min	Vit. C: 0.5% oxalic acid aqueous solution TP: ethaanol/water (50:50)		Folin–Ciocalteu reagent	Spectrophotometry with 2,6-dichlorophenolindo-phenol	[41]
90 °C and chilled	-	TAS:10-fold diluted supernatant previously centrifuged from the material TP: water/methanol (30:70)	DPPH	HPLC: diode array detector for flavonoids 256 nm, phenolic acids 325 nm, linear gradient from 20% to 80%, mobile phase A: water H_2_O: formic acid 99.8:0.2 (*v*/*v*), phase B: CH_3_OH:CH_3_CN 40:60 (*v*/*v*). A prodigy column (5 μm ODS3 100 A, 250 × 4.60)	-	[23]
64, 39–75 °C	57, 32–75 min	-	DPPH	Folin–Ciocalteu reagent	Spectrophotometry method	[24]
85 °C	30–40 min	Free phenolic fractions: methanol/water (80:20) - ultrasonic bath – ×2 supernatant with ^mol/L HCl. Bound phenolic fractions: obtained residues after phenolic extraction washed with water, blended with 4 mLL^−1^, add 6 mol/L HCl, supernatant extracted of ethyl acetate	ABTS	Folin–Ciocalteu reagent HPLC: DAD-300RS array detector, Column C18 (150 mm × 4.6 mm, 2.6 μm), mobile phase: water/acetic acid in the ratio of 99:1 (solvent A) and water/ACN/acetic acid in the ratio of 67:32:1 (solvent B), ratio flow 1 mL/min, gradient elution	-	[31]
80 °C and 90 °C	10, 20, 30 min and 10, 20, 30 min	80% ethanol	-	Folin–Ciocalteu reagent	-	[42]
80–90 °C	40–50 min	TP: Lyophilized sample extracted with methanol/0.1% formic acid (80:20). Supernatant re-extracted methanol/0.1% formic acid (80:20).	-	HPLC-MS/MS. HPLC: column C18 (3 × 75 mm, 2.7 μm), mobile phase: 0.1% formic acid (A) and acetonitrile (B), flow rate 0.6 mL/min, oven temp. 20 °C. first Gradient elution, at the end isocratic. MS: negative ionization modem source temp. 600 °C, IonSpray voltage −3500 V, nebulizing nitrogen.	-	[40]
82 °C and 85 °C	30 min and 5 min	TP and TAS: 80% methanol	DPPH	Folin–Ciocalteu reagent	-	[25]

TAS—total antioxidant status, TP—total phenolic, Vit. C—Vitamin C, CC—conventional cooking, S—steam, SV—sous-vide.

## 4. Total Antioxidant Status

Antioxidant activity determines the ability of natural or synthetic antioxidants to reduce free radicals in the tested material. Typically, polyphenols and some vitamins influence the antioxidant potential of vegetables [47]. The antioxidant potential results [Table 2] varied depending on the cooking technique, conventional cooking (CC), steaming (S) and sous-vide (SV), in relation to raw vegetables. After conventional cooking, most vegetables had a lower antioxidant potential of −4.85% (broccoli) [29] to −90.8% (carrot) [24] compared to raw vegetables, probably due to the high temperatures, causing the decomposition of compounds with antioxidant properties, as well as their transition by osmosis to “decoction” [25]. The exceptions were brussels sprouts +10.8% [29] or +20.2% (DPPH radical method) [30] and cauliflower +27.8% [29], where a higher antioxidant potential was determined compared to the raw vegetables. However, the antioxidant potential after steaming was lower compared to the raw form of seven vegetables from −2.73% (romanesco type cauliflower) [29] to −65.0% (kale leaves) [19], some of which were also characterised by higher antioxidant status (from +5.6% to +160%) by the same authors depending on the method of determining the TAS. The antioxidant potential of these vegetables determined using the DPPH radical was higher after steaming but lower according to the ABTS cation radical and FRAP methods for cauliflower [20], brussels sprouts [30], as well as kale cv crispa leaves and steam [20]. A significantly higher antioxidant potential after steaming was determined in cauliflower (+49.7%), broccoli (+39.4%) and brussels sprouts (+29.1%) [29], according to one method using the cation radical ABTS. However, an increase in antioxidant potential was found for most vegetables (6) after using the sous-vide method, and in three cases, the results were divergent depending on the author and the methods of determining the antioxidant potential. The increase in total antioxidant potential after the sous-vide process was very variable and ranged from 1.03% for kale stem [20] to 53.9% for cauliflower white rose [29]. Despite noting a lower antioxidant potential among many vegetables in relation to their raw form, these changes were lower compared to vegetables cooked by other technological processes. The exceptions were vegetables such as broccoli [25,29], cauliflower [29,30] and pumpkin [23], whose antioxidant potential was determined at a higher level after steaming, and carrots after conventional cooking compared to the sous-vide method. This may be due to the length of time of the sous-vide process compared to other culinary processes, as the extended time may contribute to a reduced antioxidant capacity. Sometimes a short culinary treatment of vegetables at high temperatures (e.g., blanching) is more beneficial than a long process [30].

A lower antioxidant potential was observed in vegetables such as grelo (72.5–88.5%) [19], col. cabdell (63.3–66.7%) [20], col. llombarda (50.0–50.3%) [21], Toscana black cabbage (64.0–64.3%) [20], artichokes (15.1–83.0%) [22], green beans (44.0–54.0%) [22], green peas (28.3–30.4% [24], carrot (28.7–90.8%) [22,24], brown teff (3.88–12.6%) [31] and white teff (1.08–30.12%). Relatively small changes were observed after the sous-vide process with only a higher antioxidant potential (40.5%) compared to the raw form observed for romanesco cauliflower [29]. After the other methods, a lower level was observed from 2.73% to 30.0%. According to Doniec and Florkiewicz [29,30], only brussels sprouts had a higher antioxidant potential determined by the ABTS cation radical method regardless of the technological process used (CC- 10.8%, S- 29.1% and SV- 4.87%), whereas the antioxidant potential of brussels sprouts determined with the DPPH radical was lower after all cooking techniques [30]. This shows how important it is to select the method for determining the antioxidant potential and properly interpret the obtained results. There was a similar discrepancy in the kale results, with a higher antioxidant potential noted for kale leaves after the SV process, and for kale steam after the S process using the DPPH radical higher from 5.60% to 44.4% and 4.60% to 160%, respectively [19]. Pumpkin also had a higher antioxidant potential after steaming than after the SV process [23]. Another example where the selection of the method for determining the antioxidant potential is important is the cauliflower white rose [20,29]. According to the TAS determinations using the DPPH radical, the antioxidant potential was higher regardless of the type of thermal treatment from 7.81% [20] to 53.9% [29] for SV, while the lower antioxidant potential was shown for the FRAP method from 40.6% [20] to 49.7% [29] for steaming.

Broccoli has been studied by Florkiewicz [29] and Özer [21] with very divergent results. The antioxidant potential determined using the cation radical ABTS was higher after technological processes than in its raw form, except for classic cooking (decrease by 4.52%), whereas the DPPH radical method indicated a lower potential after technological processes. The different results depending on the analytical methods used to determine the antioxidant potential make it impossible to analyse the changes that may occur after various technological processes.

## 5. Total Polyphenols Content

Polyphenols are a broad group of secondary plant metabolites that are divided into phenolic acids including hydroxybenzoic acids and hydroxycinnamic acids, flavonoids including flavonols, flavones, flavanones, anthocyanidins, catechins, isoflavones and chalcones, lignan and stilbenes. Their antioxidant capacity is related to the site and degree of hydroxylation in the molecule. Polyphenols shape food taste, smell, aroma and colour, as well as stabilise fats, delaying oxidative rancidity or inhibiting bacterial growth. Particular attention should be paid to the beneficial effect of polyphenols on the human body, e.g., the cardiovascular system, nervous system, eye allergy and other diseases [1,48,49,50].

As mentioned, the Folin–Ciocalteu reagent was most often used to determine the total polyphenols content, showing that most vegetables subjected to technological processes had a decreased total polyphenol content compared to their raw forms [Table 2]. The greatest loss of polyphenols was observed after the conventional cooking process from −2.17% (cauliflower white rose) [29] to −94.9% (artichokes) [22]. On the other side, vegetables total polyphenol content determined after the CC process was higher on root parsley (+7.57%) and carrot (+18.4%) [42]. There was an increase in the polyphenol content in root parsley (+5.44%) [42], brussels sprouts (+18.8%) [30], pumpkin (+218%) [23] and carrots (+17.2%) [42] after steaming. However, Florkiewicz [29] reported a smaller increase in the polyphenol content of brussels sprouts after steaming. Both authors used the F–C reagent and extracted in 80% methanol, but Florkiewicz [29] used double extraction with hydrochloric acid and acetone in the second stage, which could have resulted in a more favourable (increased amount) extraction of polyphenols [30]. A decrease in polyphenol content was observed for the other vegetables subjected to steaming, from −1.27% (cauliflower white rose) [29] to −87.5% (kale leaves) [19]. The most advantageous method was sous-vide, which resulted in the lowest reduction or increase in the polyphenols content. The polyphenol reduction ranged from −10.1% (broccoli) [22] to −86.9% (kale leaves) [19], while the increased content of polyphenols after the sous-vide was from +5.25% (broccoli) [29] but according to Özer (−13.6%) [21] to +91.0%% (pumpkin) [23]. Both authors used the F–C reagent for determinations, while Florkiewicz [29], as previously mentioned, performed a double extraction using 80% methanol, hydrochloric acid and acetone in two stages, whereas Özer [21] used only 80% methanol. Both authors used the same temperature in the sous-vide process (90 °C), while Florkiewicz used a longer time (45–50 min) [29] than Özer (5, 10 and 15 min) [21]. Elevated temperature and longer duration of the sous-vide process could have contributed to changes in the semi-permeability of protoplasmic membranes and greater polyphenol transfer during the extraction process to the extractant (methanol) [6,12].

## 6. Vitamin C Content

The vitamin C in products is not only due to its natural occurrence but also the addition of vitamin C as a food antioxidant. The content of vitamin C in vegetables is generally high (2 mg/100 g fresh carrot to 270 mg/100 g fresh parsley) [51] but degraded by factors such as pH, light, oxygen or temperature. Vitamin C is a very good antioxidant, fights free radicals by protecting DNA, reduces the risk of cancer (inhibits the formation of carcinogenic nitrosamines from nitrates and bacteria), has cardioprotective (reduces lipid peroxidation) and immunomodulatory effects (participants in the synthesis of interferon), increases the body’s immunity through the activity of T and B lymphocytes and natural killer cells and promotes the activity and transport of monocytes, granulocytes and macrophages, as well as the self-formation of selected immunoglobulins [52,53,54].

The vitamin C content was determined in 14 vegetables (Table 2) and regardless of the technological process, losses of vitamin C ranged from 1.22% to 98.3%. The lowest losses were found in asparagus after conventional treatment compared to raw vegetables [46], and the highest losses were recorded for Toscana black cabbage [19]. The sous-vide process was most beneficial to preserve the vitamin C content. In the case of eight vegetables, losses were the lowest after the sous-vide process compared to the conventional technique or steaming. This is due to vitamin C being very labile under the influence of high temperatures, the sous-vide process is performed at lower temperatures (below 100 °C) and without oxygen [20]. The lowest losses of vitamin C after sous-vide were determined in asparagus −3.66% [46] and the highest were in Toscana black cabbage −97.3% [20]. For five vegetables, conventional cooking was more beneficial than sous-vide, with higher vitamin C content after cooking at 100 °C than the sous-vide process for coll. cabdell, coll. lombarda [20], asparagus [42], pumpkin [41] and carrot [24], with losses of 90.5%, 80.0%, 1.22%, 5.01% and 17.8%, respectively, vs. 95.2%, 86.0%, 3.66%, 49.6% and 32.1%. Steam cooking was more favourable than conventional cooking for most vegetables but less favourable compared to sous-vide. Only pumpkin had the highest loss after steaming compared to other techniques including CC (55.0%) [41]. Once again, there is a difference in results among the same vegetables. According to Florkiewicz [29], cauliflower white rose subjected to steaming lost 43.4% of vitamin C (the smallest loss compared to other techniques), whereas Lafraga [19] observed the smallest loss of vitamin C for cauliflower white rose after sous-vide (97.2%). This discrepancy may be due to the different methods of vitamin C determination. Lafarga [19] determined vitamin C using HPLC, while Florkiewicz [29] spectrophotometrically. A similar situation was observed for broccoli, with Özer [21] reporting that steamed broccoli exhibited the lowest vitamin loss of 12.0%, while according to Florkiewicz [29], the sous-vide process achieved a 35.4%. These differences may also result from the use of different determination methods. Özer [21] determined the content of vitamin C by HPLC, and Florkiewicz, as mentioned earlier, spectrophotometrically [29].

The variable study results do not allow the determination of which parameters related to the technological processes and methodology of determination and/or the properties of the product (vegetable) affect the antioxidant potential and the polyphenols content. Several variables changed depending on the type of vegetable, determination methods and applied technological process, practically excluding the traditional method of classifying/grouping objects (analysed vegetables) a priori. Grouping in experimental research is the basic stage of any research procedure, and following the above, a mathematically defined similarity between objects (vegetables) was adopted as the criterion for grouping. In classical terms, this course of action is referred to as cluster analysis and is presented graphically as a cluster tree. When grouping objects, the agglomeration method was used to combine objects into successive clusters based on the similarity function (Euclidean distance), with lower-order clusters as part of the higher order. 

The grouping was performed using published data regarding the effect of conventional cooking and sous-vide on the antioxidant potential and the polyphenols content. The graphical presentation describing the effect of conventional and sous-vide cooking on the antioxidant potential (Figure 2) shows three distinct clusters (I-III), two of which (I, II) are characterised by a similar relationship of vegetables to the antioxidant potential. Cluster I includes vegetables such as carrots [22], artichokes, green beans, coll. lombarda, Toscana, coll. cabdel and broccoli [22], which were characterised by a lower antioxidant potential after the sous-vide process from −15.1% to −64.0% and after the conventional method from −50.0% to −90.8%. Additionally, the same solvent (method) and the DPPH radical method [20,22] were used in this cluster.

Cluster II includes such vegetables such as carrots [24], green peas, broccoli [21], brown teff, white teff and spinach, which were characterised by a lower antioxidant potential after technological processes than vegetables from the first cluster. Carrots [24], green peas and broccoli [21] are characterised by an antioxidant potential at the following levels: after the CC process (−28.7% to −48.5%) and after the SV process (−26.8% to −31.1%), while brown teff, white teff and spinach exhibit much lower potential losses, especially after the sous-vide process (from −1.08% to −6.60%). Most authors used methanol and the DPPH radical method, except for the determination of the antioxidant potential in brown teff and white teff vegetables using the cation radical ABTS [31]. 

Cluster III comprised cauliflower white rose [29], brussels sprouts according to two authors [29,30] (creating a weaker fifth cluster, 1–5), broccoli according to two authors [25,29] and romanesco cauliflower (forming a weaker sixth cluster-6). In all vegetables belonging to the lower order of the fifth cluster, a higher antioxidant potential was determined after technological processes: after the CC process from +10.8% to +27.8%, and after the SV process from +48.7 to +53.9%, whereas vegetables belonging to the weaker cluster six (6) were characterised by a higher antioxidant potential only after the SV process from +20.3% to +40.5% [25,29,30].

The analysis of the agglomeration chart describing the effect of conventional and sous-vide cooking on the content of polyphenols indicates two distinct clusters, among which there are differentiated lower-order clusters (Figure 3). All vegetables included in the first cluster (I) were characterised by a lower polyphenol content after technological processes. In vegetables belonging to the lower cluster of the first order (I) (coll. lombarda, grelo, artichokes), the greatest loss of polyphenols was recorded after the CC and SV processes, respectively (−83.3%, −81.8%, −94.9% and −82.7%, −84.8%, −62.5%). Vegetables such as coll. cabdell, toscana black cabbage and pumpkin [41] formed a weaker second cluster (2), and polyphenol losses after technological processes ranged from −46.7% to −64.9%. In the remaining vegetables from the first solid cluster (I), such as brown teff, carrot [22] and broccoli according to three authors [21,22,25] (the third cluster of lower orde-3), there was a fairly large reduction in polyphenols after the CC process from −34.6% to −70.4% in contrast to a small loss after the SV process from −10.2% to −40.8%. Polyphenols in all vegetables from the first cluster (I) were determined using the Folin–Ciocalteu reagent, and in most cases, methanol was used, with ethanol used in only one case [41]. The second cluster (II) of the upper order was characterised by vegetables which, after culinary processing, had a higher total content of polyphenols than their raw form, particularly after the sous-vide process. For example, the weaker fourth cluster (4), consisting of brussels sprouts [30], cauliflower white rose [29] and carrot [42], and the weaker sixth cluster (6), including vegetables such as white cardoon, broccoli, brussels sprouts and romanesco cauliflower [29], exhibited a percentage reduction in the polyphenol content from −2.17% to −46.6% after the CC process and an increase from +4.11% to +27.3% after the SV process. The fifth cluster (5) of the lower order consisted of white teff, green beans, green peas, red cardoon and carrot [24] and exhibited a lower polyphenol content after thermal processing, with the sous-vide process achieving a lower loss compared to the weaker clusters 1–3 (from −10.5% to −31.2%). In contrast, the lower-order two-element cluster (7), comprising root parsley and spinach, was characterised by a slight change in the polyphenols content in relation to their primary structure, respectively, from +3.03% to −6.02% and −6.86% to −6.00% after both CC and SV processes.

## 7. Conclusions

The published results regarding the antioxidant potential and the vitamin C and polyphenols content of vegetables after various thermal processes were very diverse, therefore it was difficult to determine which process is more beneficial. After heat treatment, vegetables such as romanesco cauliflower, pumpkin and kale steam had a higher antioxidant potential compared to their raw form. The lowest TAS reduction after classic cooking was noted for carrot and grelo, and after the SV process, brown teff, white teff, green peas and artichokes. Among vegetables, col. lombard, coll. cabdell and toscana black cabbage TAS reduction after all technological processes was at a similar level. Due to the discrepancies in the results among different authors for vegetables: brussels sprout, broccoli, cauliflower and kale leaves, it is impossible to say which process is more beneficial. Furthermore, the lack of unified research methodology or “gold standard” to determine the antioxidant potential, the polyphenols and vitamin C content of plants, which are an important source of human nutrition, limits the interpretation of the results. In addition to the considerations in the review, the authors of the publication also draw such conclusions. However, most cooking techniques contributed to reducing TAS, PC and vitamin C content, with the sous-vide process being most beneficial to reduce the losses, which is confirmed in the conclusions of other authors. Analysing the of polyphenols in order to increase their content, it can be concluded that romanesco type cauliflower and brussels sprouts should be subjected to the sous-vide process, while parsley root should be subjected to conventional processing, and pumpkin to the steaming method. The content of polyphenols in other vegetables subjected to various cooking techniques decreased. For red cardon, toscana black cabbage, artichokes, green beans, coll. cabdel, brown teff and white teff showed the smallest reduction in polyphenols after the SV process, for green peas after conventional cooking and for kale after steaming. Due to the differences in the results regarding the content of polyphenols determined by different authors, it is impossible to clearly state which technological process is more beneficial for the cauliflower vegetable. The content of vitamin C in all vegetables decreased after subjecting them to technological processes. The lowest decomposition of vitamin C was found in most vegetables after the SV method, with the exception of carrots after the CC method.

To sum up, the review of the literature available in this area shows a lack of clarity, taking into account the impact of various technological processes on their antioxidant potential, the content of vitamin C and polyphenols. In order to systematise and select the best technological process, the optimisation of the process itself and the selection of methods for determining individual components and properties of the vegetable, is necessary to create a so-called “gold standard”. The lack of a unified research methodology limits the practical aspects of interpreting the obtained test results and, consequently, interpretation difficulties, despite the use of modern methods to determine the antioxidant potential, the content of polyphenols and vitamin C in plant raw materials, which are an important source in human nutrition.

## Figures and Tables

**Figure 1 foods-12-02121-f001:**
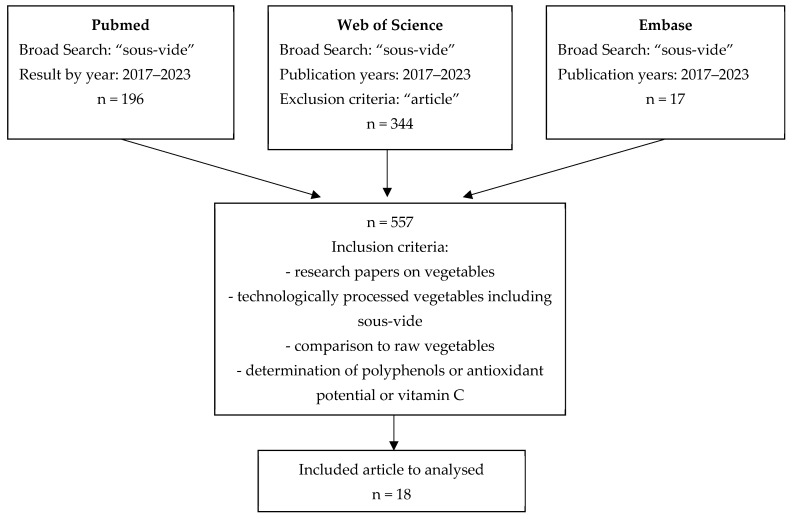
Search strategy.

**Figure 2 foods-12-02121-f002:**
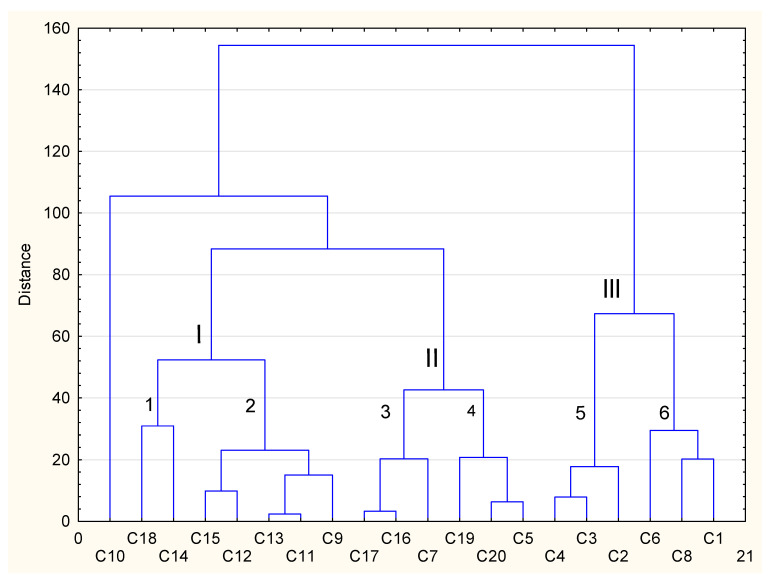
Cluster analysis of antioxidant potential of selected vegetables after conventional and sous-vide cooking in relation to their raw form. C1—romanesco type cauliflower, C2—brussels sprouts [31], C3—brussels sprouts [32], C4—cauliflower white rose [31], C5—spinach, C6—broccoli [31], C7—broccoli [21] C8—broccoli [27] C9—broccoli [24], C10—grelo, C11—coll. cabdell, C12—coll. lombarda, C13—toscana black cabbage, C14—artichokes, C15—grean beans, C_16—green peas, C17—carrot [26], C18—carrot [24], C19—brown teff and C20—white teff. I–III distinct cluster. 1–6 weaker cluster.

**Figure 3 foods-12-02121-f003:**
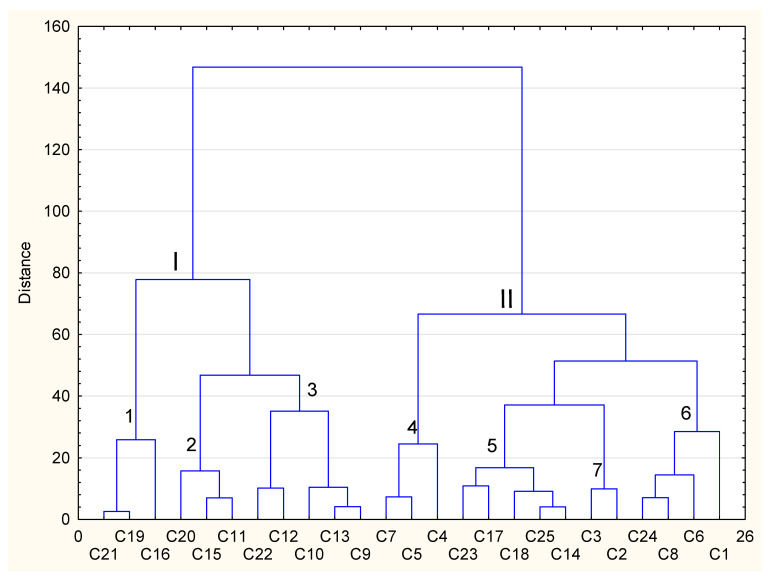
Cluster analysis of polyphenol content of selected vegetables after conventional and sous-vide cooking in relation to their raw form. C1—romanesco type cauliflower, C2—spinach, C3—root parsley, C4—carrot [44], C5—cauliflower white rose [31], C6—brussels sprouts [31], C7—brussels sprouts [32], C8—broccoli [31], C9—broccoli [23], C10—broccoli [27], C11—pumpkin [43], C12—carrot [24], C13—broccoli [24], C14—carrot [26], C15—toscana black cabbage, C16—artichokes, C17—green beans, C18—green peas, C19—grelo, C20—coll.cabdell, C21—coll. lombarda, C22—brown teff, C23—white teff, C24—white Cardoon, C25—red Cardon. I–II distinct cluster. 1–7 weaker cluster.

**Table 2 foods-12-02121-t002:** The impact of the culinary processing of vegetables on their total antioxidant status and the content of polyphenols and vitamin C compared to raw vegetables.

Vegetables	Relative to Raw Vegetables (%)									References
	TAS			TP			Vit. C			
	CC	S	SV	CC	S	SV	CC	S	SV	
Cauliflower white rose (*Brassica oleracea* var.botrytis)	↑ (27.8)	↑ (49.7)	↑ (53.9)	↓ (2.17)	↓ (1.27)	↑ (20.9)	↓ (52.7)	↓ (43.4)	↓ (46.0)	[29]
	-	↑ (40.6) DPPH method ↓ (40.2) FRAP method	↑ (7.81) DPPH method ↓ (46.0) FRAP method	-	↓ (75.0)	↓ (72.0)	-	↓ (97.7)	↓ (97.2)	[19]
Romanesco type cauliflower (green rose) (*Brassica oleracea* var.botrytis)	↓ (30.0)	↓ (2.73)	↑ (40.5)	↓ (46.6)	↓ (12.0)	↑ (7.12)	↓ (48.7)	↓ (47.1)	↓ (19.9)	[29]
Broccoli (*Brassica oleracea* var.italica)	↓ (4.52)	↑ (39.4)	↑ (34.2)	↓ (32.5)	↓ (1.87)	↑ (5.28)	↓ (58.0)	↓ (41.8)	↓ (35.4)	[29]
	↓ (48.5)	↓ (16.4)	↓ (26.8)	↓ (62.3)	↓ (8.68)	↓ (13.6)	↓ (62.3)	↓ (12.0)	↓ (36.7)	[21]
	↓ (70.0)	-	↓ (50.0)	↓ (59.9)	-	↓ (10.1)	-	-	-	[22]
	↓ (30.5)	↑ (3.13)	↑ (20.3)	↓ (70.4)	↓ (17.2)	↓ (10.2)	-	-	-	[25]
Grelo (Rapini) (*Brassica rapa* L. var rapa)	↓ (72.5)	-	↓ (88.5)	↓ (81.8)	-	↓ (84.8)	↓ (84.6)	-	↓ (81.5)	[20]
Col cabdell cv. Pastoret (*Brassica oleracea* var.capitata)	↓ (66.7)	-	↓ (63.3)	↓ (50.0)	-	↓ (46.7)	↓ (90.5)	-	↓ (95.2)	[20]
Col llombarda cv. Pastoret (*Brrascia oleracea* var.capitata f.rubra L.)	↓ (50.0)	-	↓ (50.3)	↓ (83.3)	-	↓ (82.7)	↓ (80.0)	-	↓ (86.0)	[20]
Brussels sprouts (*Brascia oleracea* var.gemmifera)	↑ (10.8)	↑ (29.1)	↑ (4.87)	↓ (18.1)	↓ (25.2)	↑ (4.11)	↓ (62.6)	↓ (52.0)	↓ (40.2)	[29]
	↑ (20.2) ABTS method ↓ (52.1) DPPH method	↑ (38.5) ABTS method ↓ (36.0) DPPH method	↑ (51.9) ABTS method ↓ (22.9) DPPH method	↓ (5.64)	↑ (18.8)	↑ (27.3)	-	-	-	[30]
Kale cv. Crispa–Leaves (*Brassica oleracea* var.acephala)	-	↑ (5.60) DPPH method ↓ (65.0) FRAP method	↑ (44.4) DPPH method ↓ (56.4) FRAP method	-	↓ (87.5)	↓ (86.9)	-	↓ (97.1)	↓ (90.0)	[19]
Kale cv. Crispa–Stem (*Brassica oleracea* var.acephala)	-	↑ (160) DPPH method ↓ (28.2) FRAP method	↑ (4.60) DPPH method ↑ (1.03) FRAP method	-	↓ (58.8)	↓ (64.7)	-	↓ (95.5)	↓ (75.9)	[19]
Toscana (black cabbage) (*Brassica oleracea* var.acephala)	↓ (64.4)	-	↓ (64.0)	↓ (62.5)	-	↓ (56.3)	↓ (98.3)	-	↓ (97.3)	[20]
Artichokes (*Cynara scolymus*, L.cv. Balnca de Tudela)	↓ (83.0)	-	↓ (15.1)	↓ (94.9)	-	↓ (62.5)	-	-	-	[22]
Green beans (*Phaseolus vulgaris* L.cv. Perona)	↓ (54.0)	-	↓ (44.0)	↓ (31.2)	-	↓ (21.2)	-	-	-	[22]
Asparagus (*Asparagus officinalis* L. cv Grande)	-	-	-	-	-	-	↓ (1.22)	↓ (11.0)	↓ (3.66)	[46]
Pumpkin (*Cucurbita moschata* cv. Leite)	-	-	-	↓ (64.9)	↓ (55.0)	↓ (49.7)	↓ (5.01)	↓ (55.0)	↓ (49.6)	[41]
	-	↑ (15.3)	↑ (14.1)	-	↑ (218)	↑ (91.0)	-	-	-	[23]
Green peas (*Pisum sativum*)	↓ (30.4)	-	↓ (28.3)	↓ (16.4)	-	↓ (17.3)	↓ (31.6)	-	↓ (30.7)	[24]
Carrot (*Dacus carota sativus*)	↓ (28.7)	-	↓ (31.1)	↓ (19.8)	-	↓ (8.84)	↓ (17.8)	-	↓ (32.1)	[24]
	-	-	-	↑ ● (18.4)	↑ ● (17.2)	↑ ▲ (23.0) ↑ ♦ (22.1)	-	-	-	[42]
	↓ (90.8)	-	↓ (44.7)	↓ (34.6)	-	↓ (40.8)	-	-	-	[22]
Root parsley (*Petreoselenium crispum* ssp. Tuberosum var. ‘Sonata’)	-	-	-	↓■ (1.51) ↑^ (7.57)	↑ ● (5.44)	↓ # (12.0) ↑ & (5.33) ↓▲ (13.8)	-	-	-	[42]
Brown teff (*Eragrostis tef* L.)	↓ (12.6)	-	↓ (3.88)	↓ (45.9)	-	↓ (35.3)	-	-	-	[32]
White teff (*Eragrostis tef* L.)	↓ (30.12)	-	↓ (1.08)	↓ (29.1)	-	↓ (10.54)	-	-	-	[31]
White Cardoon Stalks (*Cynara cardunculus* L.var.altilis DC)	-	-	-	↓ (25.5)	-	↑ (6.36)	-	-	-	[40]
Red Cardoon Stalks (*Cynara cardunculus* L.var.altilis DC)	-	-	-	↓ (22.0)	-	↓ (12.3)	-	-	-	[40]
Spinach (*Spinacia oleracea* L.)	↓ (33.2)	↓ (9.76)	↓ (6.60)	↓ (6.86)	↓ (54.6)	↓ (6.00)	-	-	-	[25]

TAS—total antioxidant status, TP—total phenolic, Vit. C—Vitamin C, CC—conventional cooking, S—steam, SV—sous-vide, ●—10, 20, 30 min, ▲—10, 20, 30 min, 90 °C, ■—10 min, ^—20, 30 min, #—10, 30 min, 80 °C, &—20 min, 80 °C, ♦—10, 20, 30 min 80 °C.

## Data Availability

Data are contained within the article.

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
