# Peer review of "The Impact of Culinary Processing, including Sous-Vide, on Polyphenols, Vitamin C Content and Antioxidant Status in Selected Vegetables—Methods and Results: A Critical Review"

_foods, 2023, doi:10.3390/foods12112121_

Round 1

Reviewer 1 Report

The impact of culinary processing, including sous-vide, on polyphenols, vitamin C content and antioxidant status in selected vegetables - methods and results: A critical review

Line number  16: You have mentioned the search years from 2017-2022 but in Figure 1, these are different. Re-check either abstract section of the figure.

Line number 21: 1st of all use the full form then use the abbreviated form of words

Line number 22: Mention the vegetables for which discrepancies were noted

Line number 25: Plant materials are linked with vegetables so try to use vegetable terms instead of plant materials as it covers a broad range

Line number 28: Background information should be mentioned in 1-2 statements

Line number 36. Please add citation of this thermal cookin staments. Please check these articls:

Effect of cooking on the nutritive quality, sensory properties and safety of lamb meat: Current challenges and future prospects. Meat Science, 175, 108172.

Enhancing the shelf stability of fresh-cut potatoes via chemical and nonthermal treatments. Journal of Food Processing and Preservation, 45(6), e15657.

Comparison of high pressure and thermal pasteurization on the quality parameters of strawberry products: a review. Food Science and Biotechnology, 32(6), 729–747.

Line number 39: It includes 7 references. These references should be in 2-3 sections.

Line number 45-46: The antioxidant potential of vegetables should be elaborated

Line number 47. Please add reference for this statement.

Line number 64: Does free radical reduction or scavenging is same?

Line number 70: Fe3+ to Fe2+: These should be in superscripts i.e., Fe3+ to Fe2+

Line number 71: Re-check the statement

Figure 1: Re-check the joining words etc.

Line number 82: Re-check F-C.

Line number 92: Citation is missing

Line number 105: Re-check citations in Table 1. There are some grammar errors as well.

Line number 135, 148: Re-check the statement

Line number 163: What is S in this statement?

Line number 171: Re-check the citations and sequencing of Table 2. The text should be justified.

Line number 183-184: The statement is quite confusing. Re-check this.

Line number 246: Mention the results

Line number 250: Mention citation for this statement

Line number 251-252: Mention the main findings/results

Line number 291: Citation is missing

Line number 313: Repetition of words

Line number 347, 369: The words are used repeatedly in both sentences. Re-check these statements

Line number 382: Volume number and page numbers are missing in some references

Good.

Author Response

Line number  16: You have mentioned the search years from 2017-2022 but in Figure 1, these are different. Re-check either abstract section of the figure.

Checked and changed for years 2017-2023.

Line number 21: 1st of all use the full form then use the abbreviated form of words

The full name and abbreviations are explained in line 10

Line number 22: Mention the vegetables for which discrepancies were noted

They were listed, for example: broccoli, cauliflower

Line number 25: Plant materials are linked with vegetables so try to use vegetable terms instead of plant materials as it covers a broad range

Corrected on plants

Line number 28: Background information should be mentioned in 1-2 statements

Background was shown on segments

Line number 36. Please add citation of this thermal cookin staments. Please check these articls:

Effect of cooking on the nutritive quality, sensory properties and safety of lamb meat: Current challenges and future prospects. Meat Science, 175, 108172.

Enhancing the shelf stability of fresh-cut potatoes via chemical and nonthermal treatments. Journal of Food Processing and Preservation, 45(6), e15657.

Comparison of high pressure and thermal pasteurization on the quality parameters of strawberry products: a review. Food Science and Biotechnology, 32(6), 729–747.

Included : Enhancing the shelf stability of fresh-cut potatoes via chemical and nonthermal treatments. Journal of Food Processing and Preservation, 45(6), e15657.

Line number 39: It includes 7 references. These references should be in 2-3 sections.

Divided into sections

Line number 45-46: The antioxidant potential of vegetables should be elaborated

Changed as suggested

Line number 47. Please add reference for this statement.

Changed as suggested

Line number 64: Does free radical reduction or scavenging is same?

They are close to specified

Line number 70: Fe3+ to Fe2+: These should be in superscripts i.e., Fe3+ to Fe2+

Corrected

Line number 71: Re-check the statement

Corrected and changed

Figure 1: Re-check the joining words etc.

Corrected and changed

Line number 82: Re-check F-C.

Checked

Line number 92: Citation is missing

Corrected and added

Line number 105: Re-check citations in Table 1. There are some grammar errors as well.

Checked and corrected

Line number 135, 148: Re-check the statement

Checked

Line number 163: What is S in this statement?

„S” is steamng - corrected

Line number 171: Re-check the citations and sequencing of Table 2. The text should be justified.

Checked and corrected

Line number 183-184: The statement is quite confusing. Re-check this.

The statement has been removed

Line number 246: Mention the results

Mention percantage results

Line number 250: Mention citation for this statement

Added

Line number 251-252: Mention the main findings/results

Mentioned the main findings/results

Line number 291: Citation is missing

Added

Line number 313: Repetition of words

Checked and corrected

Line number 347, 369: The words are used repeatedly in both sentences. Re-check these statements

Checked and corrected

Line number 382: Volume number and page numbers are missing in some references

Checked and corrected

Reviewer 2 Report

The work presented in this review is focused on an interesting subject. It deals with the differences in the polyphenols profile, Vit C and antioxidant potential found in different vegetables after cooking by different techniques (CC, S and SV), with the purpose of knowing what technique or cooking treatment should be apply for each kind of vegetal in order to preserve its beneficial properties. The main conclusions reached in this work were the big differences found by different authors, even in the same vegetal, depending on the analytical methods used for the determinations, and the general heterogeneity of the data, although it was concluded that, generally, the content of polyphenols, vitamin C and antioxidant activity were decreased after cooking, obtaining sous-vide the highest preservation.

Regardless the obtained results, there is an important lack of connection between the qualitative data presented in the table 2 and the quantitative data given in the text. By taking a look into the table 2, just a variation in terms of increase or decrease could be concluded; however, in the text, numeric data about the percentage variation are given. In this sense, in the table, is possible just to compare between different samples if the content in one of them increases and in the other decreases, but it is not possible to stablish a comparison between samples in which the content increase or decrease for both of them. Hence, I suggest to present the data (table 2) in a different way to make it easier to quantitatively compare between samples and authors.

Moreover, the following minor corrections should be made:

1. Introduction:

-        Line 33-36: Please, give the reference to this statement.

-        Line 40: Explain the term “decoction by osmosis”

2. Methods and search strategy: The figure 1 should be mentioned in the methodology description.

3. Analytical methods:

-        The table 1 collecting the different methods used for analytical determinations should be mentioned in this section.

-        Table 1: Check the general spelling. Please make clearer the description of the * and ** notes with the data collected in the table.

4. Total antioxidant status:

-        Line 118: Check, it seems to be incomplete.

-        Table 2: Check the spelling.

5. Total polyphenols content:

-        Line 194: It is said that, after CC, TP increased only in root parsley, but according to table 2 also increased in pumpkin (1/2 references) and carrot (1/3 references).

-      Line 195-196: The same for S: it is said that TP increased in root parsley and brussels sprouts, but it also increased in pumpkin and carrot.

6. Vitamin C content:

-        Line 218: Please specify the usual values of Vit C concentration found in vegetables.

-        Line 226: Check if it is 14 instead of 13.

Conclusions:

Would be interesting if the authors could say if some of the problematics that they found about obtaining strong conclusions, also were obtained by some of the authors of the included articles.

Please review the correct spelling, mainly in the tables. 

Author Response

The work presented in this review is focused on an interesting subject. It deals with the differences in the polyphenols profile, Vit C and antioxidant potential found in different vegetables after cooking by different techniques (CC, S and SV), with the purpose of knowing what technique or cooking treatment should be apply for each kind of vegetal in order to preserve its beneficial properties. The main conclusions reached in this work were the big differences found by different authors, even in the same vegetal, depending on the analytical methods used for the determinations, and the general heterogeneity of the data, although it was concluded that, generally, the content of polyphenols, vitamin C and antioxidant activity were decreased after cooking, obtaining sous-vide the highest preservation.

Regardless the obtained results, there is an important lack of connection between the qualitative data presented in the table 2 and the quantitative data given in the text. By taking a look into the table 2, just a variation in terms of increase or decrease could be concluded; however, in the text, numeric data about the percentage variation are given. In this sense, in the table, is possible just to compare between different samples if the content in one of them increases and in the other decreases, but it is not possible to stablish a comparison between samples in which the content increase or decrease for both of them. Hence, I suggest to present the data (table 2) in a different way to make it easier to quantitatively compare between samples and authors.

Thank you very much for your comments. Percentages have been added to Table 2 for better quantitative comparison between trials and authors.

Moreover, the following minor corrections should be made:

  1. Introduction:

-        Line 33-36: Please, give the reference to this statement.

Added

-        Line 40: Explain the term “decoction by osmosis”

The word entering has been changed

  1. Methods and search strategy: The figure 1 should be mentioned in the methodology description.

The figure 1 mentioned

  1. Analytical methods:

-        The table 1 collecting the different methods used for analytical determinations should be mentioned in this section.

Added

-        Table 1: Check the general spelling. Please make clearer the description of the * and ** notes with the data collected in the table.

General spelling was checked and description was cleared.

  1. Total antioxidant status:

-        Line 118: Check, it seems to be incomplete.

Checked and corrected

-        Table 2: Check the spelling.

Checked and corrected

  1. Total polyphenols content:

-        Line 194: It is said that, after CC, TP increased only in root parsley, but according to table 2 also increased in pumpkin (1/2 references) and carrot (1/3 references).

My mistake - the pumpkin was not marked by the author, but the carrot was included in the text

-      Line 195-196: The same for S: it is said that TP increased in root parsley and brussels sprouts, but it also increased in pumpkin and carrot.

Pumpkin and carrots were included and mentioned in the text.

  1. Vitamin C content:

-        Line 218: Please specify the usual values of Vit C concentration found in vegetables.

Added information about the content of vitamin C in vegetables

-        Line 226: Check if it is 14 instead of 13.

It's 14 - this has been changed

Conclusions:

Would be interesting if the authors could say if some of the problematics that they found about obtaining strong conclusions, also were obtained by some of the authors of the included articles.

Some of the authors from the included publications also made similar conclusions. This fact is included in the conclusions.

Round 2

Reviewer 1 Report

Well revised. 

Good

Reviewer 2 Report

The authors took into account all the reviewer's comments.